# Oral Reconstruction with Locoregional Flaps after Cancer Ablation: A Systematic Review of the Literature

**DOI:** 10.3390/jcm13144181

**Published:** 2024-07-17

**Authors:** Remo Accorona, Domenico Di Furia, Alice Cremasco, Luca Gazzini, Niccolò Mevio, Francesco Pilolli, Andrea Achena, Haissan Iftikhar, Shadi Awny, Giorgio Luigi Ormellese, Alberto Giulio Dragonetti, Armando De Virgilio

**Affiliations:** 1Unit of Otorhinolaryngology, ASST Grande Ospedale Metropolitano Niguarda, 20162 Milan, Italy; remo.accorona@ospedaleniguarda.it (R.A.); domenico.difuria@ospedaleniguarda.it (D.D.F.); alice.cremasco@ospedaleniguarda.it (A.C.); niccolo.mevio@ospedaleniguarda.it (N.M.); francesco.pilolli@ospedaleniguarda.it (F.P.); andrea.achena@ospedaleniguarda.it (A.A.); giorgio.ormellese@ospedaleniguarda.it (G.L.O.); alberto.dragonetti@ospedaleniguarda.it (A.G.D.); 2Department of Clinical Sciences and Community Health, Università degli Studi di Milano, 20122 Milan, Italy; 3Division of Otorhinolaryngology—Head and Neck Surgery, “San Maurizio” Hospital, 39100 Bolzano, Italy; luca.gazzini@gmail.com; 4Department of Otorhinolaryngology, University Hospitals Birmingham, Birmingham B75 7RR, UK; haissan.iftikhar@gmail.com; 5Department of Surgical Oncology, Oncology Center, Mansoura University, Mansoura 35516, Egypt; shadi.awny@gmail.com; 6Department of Organ of Sense, Sapienza University of Rome, 00185 Rome, Italy

**Keywords:** oral cancer, locoregional flaps, oral reconstruction, pedicled flaps

## Abstract

**Introduction**: The planning of oral reconstruction after tumor resection is a pivotal point for head and neck surgeons. It is mandatory to consider two aspects: the size of the surgical defect and the complexity of the oral cavity as an anatomical region. We offer a review of the literature that focuses on four types of locoregional flaps that can be profitably used for such reconstruction: infrahyoid (IF), nasolabial (NF), platysma (PF), and submental (SF). **Methods**: The study was performed according to the Preferred Reporting Items for Systematic Reviews and Meta-Analyses (PRISMA) guidelines. This systematic review was carried out according to the PICOS acronym through a comprehensive electronic search on PubMed/MEDLINE, Cochrane Library, and Google Scholar databases. For each selected article, we extrapolated eight main parameters, of which all mean values were compared through an ANOVA test. The dimensions of the oral defects were referred to as “small” (<7 cm^2^), “medium” (7–50 cm^2^), or “large” (>50 cm^2^). **Results**: A total of 139 articles were selected with a total of 5898 patients. The mean ages for each type of flap were not statistically significant (*p* = 0.30, *p* > 0.05). Seven sublocations of oral defects were reported: The most common was the tongue (2003 [34.0%] patients), followed by the floor of the mouth (1786 [30.4%]), buccal mucosa (981 [16.6%]), cheek (422 [7.2%]), hard palate (302 [5.1%]), alveolar ridge (217 [3.7%]), and retromolar trigone (187 [3.2%]). The defects were mainly medium-sized (4507 [76.4%] patients), and fewer were small-sized (1056 [17.9%]) or large-sized (335 [5.7%]). Complications were noted, the most frequent of which was flap necrosis, seen in 0.57% of cases. The functional and esthetical results were mainly positive. **Conclusions:** Locoregional flaps represent a good alternative in medium-sized defects as well as a fairly good alternative in small- and large-sized defects when other options are ruled out.

## 1. Introduction

Oral defects may depend on several causes, such as malformations [1,2,3], traumatic injuries [4], and tumor resections [5,6,7]. The latter is of special interest for head and neck surgeons, as it poses peculiar dilemmas. Especially in the treatment of malignant tumors, the main target remains radical resection [8,9]. This has led to the necessity of refining the reconstructive side of oncological surgery in the treatment of oral carcinomas [9,10,11,12]. In planning oral reconstruction, it is mandatory to consider two aspects: on one hand, the size of the surgical defect; on the other hand, the complexity of the oral cavity as an anatomical region, which identifies several subsites with various implications in terms of function and cosmesis [10,11,12]. The gold standard of oral reconstruction is, up to now, the free flap transfer; however, for selected patients unfit for free flap reconstruction, or in the case of specific defects, the reconstructive surgeon needs to have a wide armamentarium of alternatives. The aim of the present paper is to present the most common locoregional flaps used in this regard. We specifically offer a review of the literature that focuses on four types of locoregional flaps: IF, NF, PF, and SF.

## 2. Methods

The study was performed according to the Preferred Reporting Items for Systematic Reviews and Meta-Analyses (PRISMA) guidelines [13]. Institutional review board approval and informed consent were not required for this review of previously published studies. No review protocol was registered for this study. 

### 2.1. Eligibility Criteria

This systematic review was carried out according to the PICOS acronym: Patients (P), adults that underwent oral oncological reconstruction with submental flap, or infrahyoid flap, or platysma flap, or nasolabial flap; Intervention (I) oral oncological reconstruction with submental flap, or infrahyoid flap, or platysma flap, or nasolabial flap; Comparison (C), not necessary; Outcomes (O), both functional and cosmetic outcomes; Study design (S), retrospective cohort studies. 

Exclusion criteria were as follows: non-English language; insufficient reported data; not-extractable data; cohorts including less than two patients by type of approach, oral defect non-generated after tumor resection, conference abstract, letter to the editor, book chapter, case reports, or technical notes. 

### 2.2. Data Source and Study Searching

A comprehensive electronic search was performed in PubMed/MEDLINE, Cochrane Library, and Google Scholar databases. Relevant keywords, phrases, and MeSH terms were adjusted to fit the specific requirements for each of the individual databases. An example of a search strategy was the one used for PubMed/MEDLINE: oral reconstruction AND infrahyoid flap; AND nasolabial flap; AND platysma flap; AND submental flap. The “cited by” function on Google Scholar was used to obtain other relevant articles for the study. Finally, a cross-reference search of the selected articles was performed. The last research was conducted on 31 December 2022.

For each selected article, we extrapolated eight main parameters: type of flap, number of patients involved in each study, mean age, whether they were fit or not for a free flap, the location of the defect, its dimension, early complications, and functional/esthetical results. All the mean values were compared through an ANOVA test.

As previously assessed, we focused on four types of flaps.

### 2.3. Infrahyoid Flap

This flap was first described by Wang and Shen in 1980 [14] when it was mainly used for the reconstruction of oral defects following tongue resections, evolving from a simple muscle transposition to a versatile pedicled myocutaneous flap. As for today, it is also used for the reconstruction of defects involving the oral floor, the oropharynx, the hypopharynx, the parotid area, and the lower face, which was possible thanks to further modifications that improved venous return problems and donor site aesthetics [15,16,17]. Its vascularization depends on the superior thyroid pedicle, while the innervation comes from the ansa cervicalis [18]; it includes the upper parts of the sternohyoid, the sternothyroid, and the omohyoid muscles [12,19]. The size of the skin pad above the muscular plan can be adjusted, with the possibility of reaching up to 9 cm in length and 5 cm in width [12]; however, this flap proved to be much better in the reconstruction of small- and medium-sized [12] than in large defects [20]. Typically, the flap is planned so that the medial margin falls along the midline, the lateral margin remains 3 to 5 cm to the medial one, the upper margin follows the line of the hyoid bone, and the lower margin coincides with the suprasternal notch [18]. The harvesting of the flap continues through to its deep plane, that is the thyroid fascia; usually, its preparation begins on the medial margin, to end on the lateral margin, where the pedicle can be isolated [18]. Whenever the defect of the floor of the mouth is full-thickness, the flap can be transposed using a pull-through technique; otherwise, an artificial channel can be created [18].

### 2.4. Nasolabial Flap

The use of the nasolabial flap dates back to 600BC but has been widely upgraded throughout the centuries and especially in recent times: The first description in modern times dates back to 1974, when the transposition of just the infrahyoid muscle was used to repair the anterior floor of the mouth, followed by further modifications made in 1984, when the overlying skin was added to assure better aesthetical results [21]. It is characterized by two different vascular pedicles depending on the subtype of flap is to be used: The inferiorly based nasolabial flap depends on the facial artery while the superiorly based nasolabial flap depends on the infraorbital and transverse facial arteries [9]. This flap offers the advantage of providing a hairless tissue [9], and its size allows an adequate coverage of small- and medium-sized defects [22,23]. One peculiar aspect of this flap is that it is normally used to reconstruct contralateral defects, while its use for ipsilateral defects is still controversial [24]. The flap is usually drawn as a fusiform 2.5–4 cm figure, where the upper tip is located roughly 5 mm under the medial canthus, the lower tip 1.5 cm lateral to the oral commissure, and the medial border follows the nasolabial fold [25]. Once the flap is completely harvested, it is rotated and tunneled through the muscles of the floor of the mouth to reach the defect [25].

### 2.5. Platysma Flap

The platysma flap was first described in 1969 by Farr et al. [26] and has undergone several changes throughout the years, going from a simple muscle transposition to an actual flap where the overlying skin was added to assure an adequate aesthetical result [27,28,29]. This flap, too, can have two different blood supplies depending on the subtype of flap applied: The inferiorly based flap uses the transverse cervical artery, but it has no application for head and neck reconstruction [30]; the superiorly and posteriorly based flap uses the submental artery [31], which is a branch of the facial artery. The type of rotation this flap offers is especially useful in the reconstruction of defects regarding the anterior and lateral floor of the mouth, retromolar trigone, and parotid region [32], and serves its purpose for small-, medium-, and even some large-sized defects [33,34,35]. The flap can be harvested in three different ways. In the apron incision, the inferior border is at the clavicle level, the anterior margin starts at the chin medially, and the posterior incision starts at the mastoid process; a horizontal elliptic flap is obtained [28,35]. The MacFee incision, which consists of a submandibular incision, also allows for obtaining a horizontal elliptic flap [29,35]. With the T-shaped incision, a vertical incision is performed in the middle of the pedicle, allowing for a vertical elliptic flap [27,35].

### 2.6. Submental Flap

The submental flap, first described in 1993 by Martin et al. [36], is one of the most popular flaps used to reconstruct small- and medium-sized defects regarding the anterior part of the mouth and of the mandible; this flap was specifically designed to solve a series of problems linked to the other locoregional flaps, notably the length of the pedicle and the arch of rotation. A breakthrough in this respect was made in 2007, when an alternative technique with early vessel dissection was introduced, allowing for an even longer pedicle with higher motility [37,38]. Its blood supply derives from the largest branch of the facial artery emerging right above the submandibular gland [39,40]. The harvesting of this flap begins with outlining an elliptical skin island in the submental area, as large as it is required to cover the defect, but not so much as to prevent the primary closing of the incision. A horizontal incision is then extended to the angle of the mandible; at this site, it is important to isolate and preserve the marginal mandibular nerve to allow for the identification of the facial artery and vein, which are to be carefully separated from the submandibular gland. The anterior belly of the digastric muscle is thus detached from the symphysis menti. The flap is finally tunneled through the mylohyoid muscle, following Wharton’s duct [41].

### 2.7. Dimensional Criteria

The dimension of the oral defect following tumoral resection was referred to as “small”, “medium”, or “large” in the articles, often without a reference to the actual dimensions in centimeters. Furthermore, the dimension in centimeters varied widely among all of the articles. Following the most frequent trend in literature, we defined “small” as those defects <3 cm, “medium” as those between 3 and 8 cm, and “large” as those >8 cm.

## 3. Results

A total of 139 articles were selected, according to the inclusion and exclusion criteria we set a priori (Figure 1). A total of 18 regarded the IF, 34 the NF, 32 the PF, and 55 the SF. A total of 5898 patients were thus taken into consideration. The mean ages for each type of flap were recorded in a similar fashion, resulting in a mean of 58.4 for the IF, 58.6 for the NF, 56.7 for the PF, and 58.5 for the SF; the difference in mean ages between the groups was not statistically significant (*p* = 0.30, *p* > 0.05). The patients’ fitness for the alternative use of a free flap was not always indicated in the articles we selected but was otherwise noted; patients were thus eligible for an alternative free flap in 29.4% of cases in the IF, in 25.0% of cases in the NF, in 42.8% of cases in the PF, and in 23.7% of cases in the SF. Seven sublocations of oral defects were reported in the articles: We found 2003 patients having tongue defects, 1786 with floor of the mouth defects, 217 with alveolar ridge defects, 187 with retromolar trigone defects, 981 with buccal mucosa defects, 302 with palatal defects, and 422 with cheek defects (Table 1). The articles graded the defects as small, medium, and large sizes. Small-sized defects were fairly frequently addressed, with a total of 1056 patients. All the flaps were mainly used to repair medium-sizes defects, with a total of 4507 patients. Large-sized defects were only involved in a minority of cases, with 335 patients (Table 2). Seven main early complications were observed in the various studies: Flap necrosis was the most frequent, seen in 34 cases (0.57%); the second was flap loss, seen in 27 cases (0.45%); dehiscence of the donor site was noted in 23 cases (0.39%), followed by orocutaneous fistulas, which occurred in 21 cases (0.35%); skin necrosis was also noted fairly often, with a total of 17 cases (0.29%); and the least frequent complications were oral incompetence, only seen in two cases (0.03%), both in the NF, and trismus, seen merely in one case in an NF study. We report in Table 3 the flaps that encountered the most complications. Functional and esthetical results after operation were rarely taken into considerations by the authors throughout the years. In 66 cases (1.12%), oral diet was considered viable, while in just three cases (0.05%), it was deemed impossible and required alternative solutions. The cosmetic results were deemed as good in 57 cases (0.97%) and bad in just six cases (0.10%), based on the patient subjective perspective. Verbalization was noted adequate in 37 cases (0.63%), while the other articles did not specify this parameter (Table 4). All articles pointed out how the positioning of a feeding tube following surgery is mandatory to allow for correct healing of the wound and the flap. The mean span to restore normal oral diet was of 13.6 days for the IF, 9.5 days for the NF, 13.6 days for the PF, and 9.6 days for the SF. The three cases where oral diet was considered impossible needed to resort to Percutaneous Endoscopic Gastrostomy.

## 4. Discussion

When dealing with surgical reconstruction after cancer resection in the oral cavity, the first concern should be the assessment of the obtained defect size. Small-sized defects can often be resolved through a primary closure or a local intraoral flap, while medium-/large-sized defects very often necessitate reconstructive surgery to obtain an acceptable result [11,42,43,44]. This involves different types of flaps, either local, locoregional, distant pedicled, or free flaps. Local flaps are extensively useful for small- and medium-sized defects as they usually provide the same type of tissue that is removed; they are quick to harvest and they do not require specific skills to be performed [10,45,46]. However, they present with some limitations, such as their limited excursion and their inapplicability to composite defect reconstructions [43,47]. Free flaps, especially the radial forearm, the anterolateral thigh, and the fibula flap [48], are the optimal types of flaps to be used because of their versatility and reliability [12,49,50,51]. Despite this, their usage is usually limited as they require specific microvascular surgery skills and thus cannot be performed in every institute [8]. Considering the pros and cons of every type of flap, the locoregional flaps represent a good compromise as they contribute with a larger excursion while being easier to harvest as they don’t require specific skills in reconstructive microsurgery [8]. Locoregional flaps have mainly been used to repair medium-sized defects in the oral compartment [10,14,21,26,36,45,46]. Our study aimed to offer a comprehensive review of the literature to properly assess the actual indications for four specific locoregional flaps in repairing different sizes and locations of oral defects. The first analysis was conducted on the different age spans included in each article to see if age could be a factor that would favor the use of a specific flap; however, no statistically significant differences were noted among the four groups, which proved that the four flaps can be used interchangeably at all ages. Secondly, we noted the patients’ comorbidities in terms of their eligibility to also undergo a free flap harvesting. Notably, local flaps like the ones we focused on are used in all patients where a free flap is not feasible due to old age or comorbidities [52]. Our review partially confirmed this aspect as the majority of articles included patients who could not bear the long operations needed to harvest a free flap. However, it is interesting to acknowledge that a great number of studies included patients who underwent locoregional flap harvesting despite the absence of comorbidities that would define them as unfit for a free flap. This is specifically possible due to the similarity of the overall results of these two techniques, leading to the surgeons’ preference for the easier and faster locoregional flap harvesting rather than the free flap approach [17,35,53,54,55,56,57,58,59,60,61,62,63,64,65,66,67,68,69,70,71,72,73,74,75,76,77,78,79]. Once these preliminary aspects were ruled out, the four flaps were then compared based on two main technical aspects: the size of the defect they had to cover and its location. As mentioned prior, the main issue we encountered regarding the size of defect was that no unanimous measures were used to define small, medium or large defects: most articles merely used the terms “small”, “medium”, or “large” without further details [17,80,81,82], while others used measures that vastly differed from one another [12,16,18,19,63,76,83,84,85,86,87,88]. For such reasons, after careful comparison between all articles, we established specific measures to the differentiate the sizes following the most common trends: small (<3 cm), medium (3–8 cm), and large (>8 cm) [86]. Secondly, we identified seven main locations often mentioned in tumoral oral resection descriptions: tongue, floor of the mouth, alveolar ridge, retromolar trigone, buccal mucosa, cheek, and palate [52,65]. To the best of our knowledge, no one in the literature ever attempted to give a comprehensive algorithm based on these two aspects, and such is our aim. Whenever an oral tumoral resection results in a small defect—smaller than 3 cm—primary or secondary closure should always be taken into consideration first [11], depending on the specific location involved. However, other options may be considered within this size range, providing similar results: the SF may be used in tongue, floor of the mouth, buccal mucosa, palate, and cheek defects; the NF may find its use in tongue and retromolar trigone defects; and the PF might be resourceful in floor of the mouth, alveolar ridge, retromolar trigone, and buccal mucosa defects. Medium defects—between 3 and 8 cm in width—are the ones most prone to the use of locoregional flaps in accordance with the literature [10,45,46]. The SF appears to be the most versatile among the four flaps we studied as it can be used for the reconstruction in all locations, with the sole exception of the retromolar trigone, where other options seem to be more adequate according to our analysis. The PF proved useful in the reconstruction of medium defects in the floor of the mouth, alveolar ridge, retromolar trigone, and buccal mucosa. The NF finds its use in the reconstruction of the alveolar ridge and the retromolar trigone. The IF seems to be the least versatile as it is only the most profitable in the reconstruction of the floor of the mouth and tongue. Large defects measuring more than 8 cm should always be reconstructed with free flaps whenever possible [12,49,50,51]. There are some cases, however, where patients are unfit for them: inadequate vascular reservoir [89], coexisting systemic diseases, or an American Society of Anesthesiologists Physical Status (ASA PS) classification of 3–4 [90]. Pedicled flaps like the Pectoralis major muscle flap or locoregional flaps represent a concrete alternative in these situations, with SF being the main choice in all the oral subsites. PF may also be used in the reconstruction of the retromolar trigone (Figure 2a,b). 

The possibility of freely using locoregional flaps also comes from their reliability and safety. As we found out in our review, serious complications are extremely scarce, with just 2.12% of the 5898 patients studied. Furthermore, only 1.03% of these cases resulted in actual flap necrosis or loss. Moreover, it is important to note that the articles never linked the insurgence of complications to the size of the defect, which might instead prove a crucial point given that large defects may be more prone to complications. Additionally, most of the studies described excellent masticatory function, cosmetic results, and verbalization after their use, with just 0.05% of cases that required a permanent gastrostomy and 0.10% of cases that were not satisfied with the obtained cosmetic results. The oral diet parameter deserves a special note: We found that a relatively short amount of time was required to switch from the feeding tube to normal oral diet when using a locoregional flap (with a mean of 13.6 days for the IF, 9.5 days for the NF, 13.6 days for the PF, and 9.6 days for the SF). This is especially outstanding when compared to the same parameter using a free flap (26.5 days according to Sittitrai et al. [74]). However impressive these results may appear, it is important to note that very few of the articles analyzed actually used this parameter (77.77% of the articles for the IF, 11.76% of those for the NF, 15.63% of the papers for the PF, and 27.27% of those for the SF). Thus, a surgeon that faces the problem of reconstruction after oral oncological resection needs to be aware that clinical results might be tricky, especially in the immediate post-operative time. However, all possible complications described in the articles should and can be easily addressed. On the one hand, patients that end up with permanent gastrostomy should immediately be addressed for logopedic trainings, which might prove useful to restart correct deglutition. As for flap necrosis or loss and bad aesthetic results, the relative advantage of locoregional flaps come into place: in both cases, there would be a need for reintervention. If the reconstruction started with a more advanced flap (i.e., a pediculed or free flap), the reintervention might prove challenging, as following flaps might not be feasible anymore, or they would have to be harvested on an already compromised tissue. On the contrary, starting with a locoregional flap does not affect the use of a more complicated flap that needs to be harvested as a second intention.

In these respects, our systematic review aims to provide a comprehensive look at the different options surgeons have in order to face complicated situations in which a clear direction to be taken for reconstruction is hard to identify, based on the experience of several surgeons throughout the years. However, the main aim of our study also reveals one of its greatest limitations as well as a general limitation for all the articles included in the study itself: A clear decision-making algorithm has never been made for these reconstructions, and as a result, each surgeon arbitrarily decided to use one type of flap over another, solely basing the decision on personal skills and knowledge. Thus, it is difficult to state that certain flaps absolutely cannot be used for certain defect sizes or locations not included in our paper. Furthermore, like stated prior, complication rates and the functional and aesthetical results are very rarely related to secondary aspects other than the type of flap used (i.e., clinical conditions of the patient, type of tumor, and tissue conditions after resection). It is thus difficult to foresee whether the same flap might have two very different results on two patients that started from different conditions. It is our advice to always take these aspects into consideration in order to assure the best results for the patient. The application of our algorithm to future cases of reconstruction might also help future publications in this respect, giving a standardized schema for everyone to follow. Additionally, further and deeper focus on the relation between complications rates, functional results and, secondary aspects that might affect them is warranted in future studies. In our opinion, a higher focus on this aspect should be made in the future, as this represents one of the main post-operative goals of recovery and thus a more accurate comparison with other types of flaps might help the surgeon choose wisely.

## 5. Conclusions

The reconstruction of defects in the oral cavity after tumoral resection is a multifactorial problem that requires patient-tailored solutions to give the best cosmetic and functional results. Locoregional flaps represent the perfect option in medium-sized defects, as well as a fairly good alternative in small- and large-sized defects, when other options are ruled out. We hereby propose an easy-to-follow algorithm to help head and neck surgeons in making a rational choice of the flap to use in various situations, and we suggest further studies to be made for validation.

## Figures and Tables

**Figure 1 jcm-13-04181-f001:**
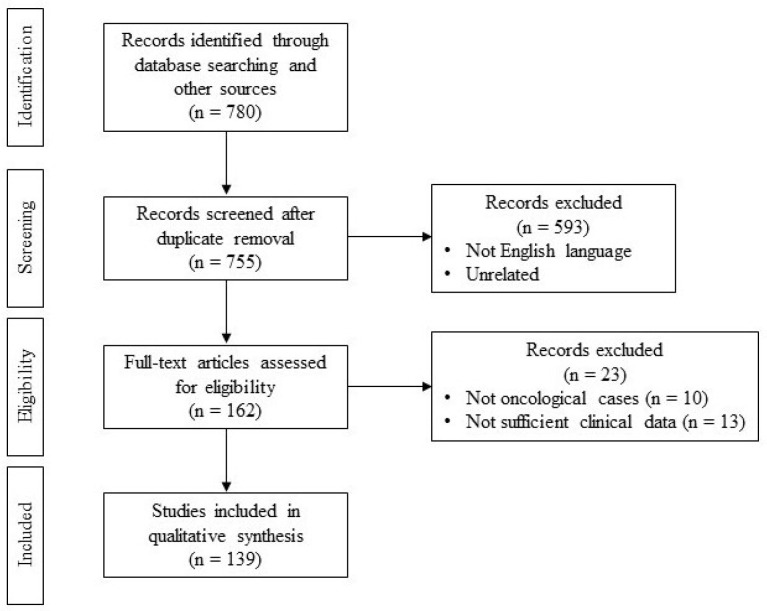
PICOS diagram.

**Figure 2 jcm-13-04181-f002:**
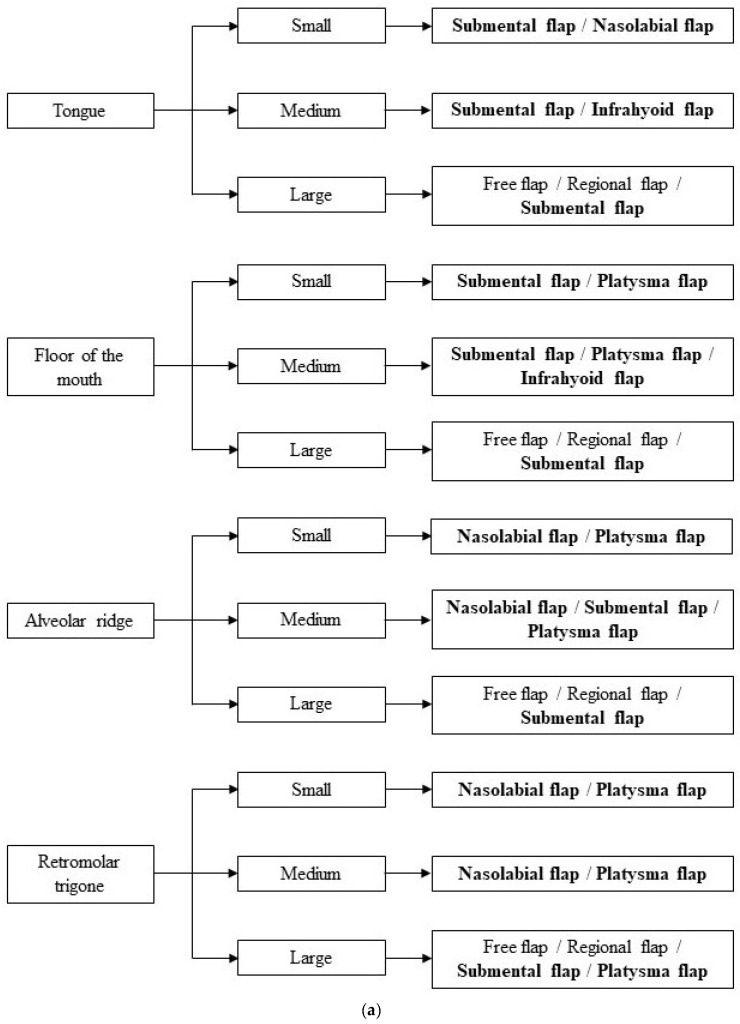
(**a**,**b**) Surgical reconstruction choice based on subsite of the oral cavity and defect size.

**Table 1 jcm-13-04181-t001:** Number of patients per defect location.

	Infrahyoid Flap	Nasolabial Flap	Plastysma Flap	Submental Flap
Tongue	380 (52.9%)	219 (21.4%)	199 (18.5%)	1205 (39%)
Floor of the mouth	275 (38.3%)	372 (36.3%)	437 (40.7%)	702 (22.8%)
Alveolar ridge	16 (2.2%)	95 (9.3%)	45 (41.9%)	61 (2%)
Retromolar trigone	3 (0.4%)	95 (9.3%)	60 (5.7%)	29 (1%)
Buccal mucosa	44 (6.1%)	220 (21.5%)	278 (25.9%)	439 (14.2%)
Palate	0 (0%)	21 (2%)	7 (0.6%)	274 (8.9%)
Cheek	0 (0%)	0 (0%)	48 (4.5%)	374 (12.1%)

**Table 2 jcm-13-04181-t002:** Number of patients per dimension of defect.

	Infrahyoid Flap	Nasolabial Flap	Platysma Flap	Submental Flap
Small	103 (14.3%)	380 (37.2%)	369 (34.4%)	204 (6.6%)
Medium	615 (85.7%)	605 (59.2%)	659 (61.4%)	2628 (85.2%)
Large	0	37 (3.6%)	46 (4.2%)	252 (8.2%)

**Table 3 jcm-13-04181-t003:** Complications.

	Infrahyoid Flap	Nasolabial Flap	Plastysma Flap	Submental Flap
Flap necrosis (0.57%)	6	4	11	13
Flap loss (0.45%)	1	6	6	14
Dehiscence of the donor site (0.39%)	0	9	5	9
Orocutaneous fistulas (0.35%)	1	4	6	10
Skin necrosis (0.29%)	6	0	7	4
Oral incompetence (0.03%)	0	2	0	0
Trismus	0	1	0	0

**Table 4 jcm-13-04181-t004:** Functional and esthetical results.

	Infrahyoid Flap	Nasolabial Flap	Plastysma Flap	Submental Flap
Viable oral diet (1.12%)	12	18	12	24
Non-viable oral diet (0.05%)	0	2	1	0
Good cosmetic results (0.97%)	2	18	12	25
Bad cosmetic results (0.10%)	0	3	1	2
Adequate verbalization (0.63%)	2	9	4	22

## Data Availability

Not applicable.

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
