# Peer review of "Oral Reconstruction with Locoregional Flaps after Cancer Ablation: A Systematic Review of the Literature"

_jcm, 2024, doi:10.3390/jcm13144181_

Round 1

Reviewer 1 Report

Comments and Suggestions for Authors

The authors conducted a review presenting different local flaps in order to reconstruct the oral cavity. Methods are clear, literature is adequate. Grammar and style are also sufficient.

However, it remains unclear how the submental flap cand be used for cover of palatal defects, also the coverage of trigonum retromolare with nasolabial flap is interesting to be described in a case example.

The idea in categorizing small, medium and large defects works very well. Figures should be resubmitted in better image quality. Results should comprise percentages for better understanding. 

Comments on the Quality of English Language

Adequate.

Author Response

The authors conducted a review presenting different local flaps in order to reconstruct the oral cavity. Methods are clear, literature is adequate. Grammar and style are also sufficient.However, it remains unclear how the submental flap cand be used for cover of palatal defects, also the coverage of trigonum retromolare with nasolabial flap is interesting to be described in a case example.

Response: The submental flap can be used to cover palatal defects: once the flap has been harvested, a submucous tunnel is created between the vestibular borders, and to extra oral incision; the submental flap is hence brought to the oral cavity through this tunnel, and is anchored to the intact hard palate with bone sutures. Such technique is already described elsewhere in literature, and we decided not include it in our articles, as mainly focuses on choosing the right flap in the right area, and it doesn't mean to indicate the specific way of repairing each of the numerous areas.

Figures should be resubmitted in better image quality.Results should comprise percentages for better understanding.  

Response: We will update the paper with better images quality and percentages included in the tables.

Reviewer 2 Report

Comments and Suggestions for Authors

Thank you for the opportunity to review this paper:

The following are some points about each section that may prove to be useful to the reader:

The background demonstrated in the introduction is accurate yet it can be imperative to expand on the historical advancement of each type of flap to establish a scope of the work. 

In the article, you described the applied methodology in detail; however, to optimize the understanding of the flow of the study, I believe that it would be helpful to include a flowchart to explain the selection of the studies, which might look like a PRISMA flowchart. 

The details are described in a manner that is easily understandable and precise. Nevertheless, it would be beneficial when the author include a summary table that contrasts the primary end points (for instance, complication rates, or functionality) of the various flap types. 

The discussion is useful but may be more robust with a focus on the possible biases and limitations within the included papers. Also, there must be benefits in discussing possible directions of future publications. 

The conclusion allows for summarizing all the discovered truths and overall discussion of the question. There was some discussion of implications of the results from a clinical perspective but it could have been expanded on and made more clearly applicable to the practitioner. 

All in all, this paper is a systematic review of reliable quality and high importance for the field of oral reconstruction of patients undergoing cancer ablation. The information into which they are organised is dissected with sensible precision, and the investigation is solid. 

Author Response

-Imperative to expand on the historical advancement of each type of flap to establish a scope of the work.

Response: ok

-Flowchart to explain the selection of the studies, which might look like a PRISMA flowchart. 

Response: We will update the new version with the Prisma flowchart.

-Summary table that contrasts the primary end points (for instance, complication rates, or functionality of the various flap types.

Response: We already showed two tables: one with with the complication rates and the other one with aesthetical and functional result for each flap (we will however add the percentages for better clarification).

-The discussion is useful but may be more robust with a focus on the possible biases and limitations within the included papers. Also, there must be benefits in discussing possible directions of future publications. 

Response: Ok

-There was some discussion of implications of the results from a clinical perspective but it could have been expanded on and made more clearly applicable to the practitioner. 

Response: Ok